# Parental Factors Related to Adolescent Girls’ Suicide Attempts: A Cross-Sectional Study from 2015 to 2018

**DOI:** 10.3390/ijerph18158122

**Published:** 2021-07-31

**Authors:** Yeon-Jung Lee, Jin-Young Lee, Minjae Kim

**Affiliations:** 1Department of Psychiatry, Soonchunhyang University Seoul Hospital, College of Medicine, Soonchunhyang University, Seoul 04401, Korea; joa.young424@gmail.com; 2Department of Applied Statistics, Chung-Ang University, Seoul 06911, Korea; 109154@schmc.ac.kr

**Keywords:** adolescent, mothers, suicidal ideation, suicide, risk factors

## Abstract

Adolescent suicide is a serious global health concern. Although familial transmission of suicidal behaviors has been identified in previous research, the effects of parental gender remain unknown. This study identified the influence of parental suicidal behaviors on suicide attempts among adolescent girls. We collected data through a cross-sectional, nationwide, population-based survey in South Korea and evaluated data from 890 adolescent girls (aged 12–18 years) who had attempted suicide and their parents. Hierarchical logistic regression was used to analyze the risk factors related to suicide attempts among adolescent girls. The final model indicated that mothers’ suicidal plans and attempts (*OR* = 6.39, *OR* = 12.38, respectively) were important risk factors for suicide attempts in adolescent girls. Future studies should identify specific methods for effective prevention and treatment through path analysis of the related factors affecting suicidal behavior of adolescents according to their parents’ gender.

## 1. Introduction

Adolescent death by suicide is a serious social concern [1]. The World Health Organization reported that suicide is the second leading cause of death among youth aged 10–19 years [2]. Moreover, South Korea has a high suicide rate, and it is the most common cause of adolescent (10–19 years) deaths [3]. The adolescent suicide rate is reported to be higher in boys than in girls. However, recent reports suggest that the suicide rate among girls has increased and that the gender gap in suicide is narrowing [2,4]. In the United States of America, the average suicide rate among girls aged 10–14 years has increased by nearly 13% per year, compared to 7% for boys, since 2007 [4]. In South Korea, the teenage suicide rate in 2019 increased by 38.6%, compared to 2015. While the suicide rate for adolescent boys increased by 21.3% (up 1 per 100,000 population), the suicide rate for adolescent girls increased by 69.2% (up 2.7 per 100,000 population) [3]. From 2017 to 2019, the suicide rate for adolescent boys decreased by 5.5% (down 0.3 per 100,000 population), although for adolescent girls it increased by 82.9% (up 2.9 per 100,000 population) [3]. To prevent an increase in the suicide rate of adolescent girls, it is necessary to identify associated risk factors and develop therapeutic interventions.

Risk factors for adolescent suicide can be divided into internal individual and external environmental factors. While internal individual factors include gender, sleep deprivation, previous suicidal behaviors, genetic factors, history of mental illness, and physical disorders, external environmental factors may include peer relationships, familial relationships, school bullying, and stressful life events [5,6,7,8]. In addition, the risk factors leading to suicide are diverse and the internal individual and external environmental factors interact with one another in a complex manner [5,7]. However, most studies report common risk factors for both genders, while some have examined the gender paradox, which refers to the idea that girls attempt suicide more often than boys, yet boys have higher suicide rates and use more fatal methods than girls [6,9,10]. Prior studies report that, after controlling for common risk factors, the suicide risk was higher in adolescent girls with depression or internalizing problems [8,10] and in adolescent boys with conduct disorders [11]. Adolescent girls who died by suicide had a higher rate of depressive symptoms and a history of self-injury and suicide attempts, compared with adolescent boys [12]. Other studies have consistently reported that adolescent girls have a higher rate of self-injurious behavior and suicide attempts than adolescent boys [6,13,14]. Moreover, suicidal attempt history was reported to be strongly associated with subsequent completed suicide compared to other risk factors in previous studies [8,15,16]. Therefore, identifying the risk factors associated with suicide attempts among adolescent girls may help in lowering suicide rates.

In previous studies [8,17,18], suicidal behaviors, including suicide attempts, had high familial transmission, which is a risk factor for increasing suicide rates among adolescents. Furthermore, as studies have revealed the direct effect of parental suicidal behaviors on adolescents through heredity, parental suicidal behaviors are internal individual risk factors. In addition, parents’ poor problem-solving ability and lack of communication skills can act as external environmental risk factors for adolescents’ suicidal risk, while also resulting in marital and child-parent conflicts [19,20]. In previous studies, the effect of parental mental health differed depending on the adolescents’ gender, especially for adolescent girls, who were more affected by maternal mental health [21,22,23,24]. From 2017 to 2019, the adolescent suicide rate in South Korea increased by 18.4%. Moreover, that of their parents’ generation, those in their 40s and 50s, also increased by approximately 10% (11.1% and 8.1% increases among those in their 40s and 50s, respectively) from 2017 to 2019. In addition, the suicide rate of female adults increased by 23.5% and 16.1% in their 40s and 50s, respectively, from 2017 to 2019; this finding may be related to the increased adolescent girls’ suicide rate (82.9%). To the best of our knowledge, there have been no studies evaluating differences in the risk factors of suicide attempts among adolescent girls according to their parents’ gender.

Considering the increasing suicide rates among adolescent girls in South Korea and the effect of parents’ suicidal behavior on their offspring, we investigated the risk factors related to suicidal attempts, and the influence of parental suicidal behaviors on suicidal attempts of adolescent girls.

## 2. Methods

### 2.1. Participants

This study was conducted using data gathered from the Korean National Health and Nutrition Examination Survey (KNHANES) by the Division of Chronic Disease Surveillance of the Korean Centers for Disease Control and Prevention. We analyzed data from 2015 to 2018 [25,26] using a cross-sectional design. Since the KNHANES launched a questionnaire on suicidal behaviors of adults and adolescents in 2015, this study analyzed data from 2015 to 2018 to confirm the relationship between suicidal behaviors of adolescent girls and their parents.

The participants comprised 890 adolescent girls aged 12–18 years and their 1500 parents (645 fathers, 855 mothers); they were representative of a nationwide non-institutionalized population in South Korea [26]. The Institutional Review Board/Ethics Committee of Soonchunhyang University, Seoul Hospital of Korea approved the study protocol (IRB No. 2021-01-002). The data used are publicly available from the Korean Centers for Disease Control and Prevention (2018). Informed consent was not required in this study as the data were retrieved from a publicly available database.

### 2.2. Procedure

In this study, we used data from a health interview (demographic, social, health, and nutritional status), health examination, and nutrition survey of participants collected in the KNHANES [26]. The health interview consisted of demographic characteristics and clinical factors. A trained interviewer evaluated the demographic characteristics through face-to-face interviews with participants at a mobile examination center (MEC). Participants’ clinical factors were assessed using a self-reporting scale in the MEC [26].

### 2.3. Study Variables

In this study, we investigated sociodemographic factors (i.e., sex, age, socioeconomic status, parents’ education, and adolescent girls’ sleep duration) and mental health factors (i.e., stress, depressed mood, suicidal ideation, suicidal plan, counseling experience, and quality of life). The average sleep time per day was assessed using the open-ended question, “What is your average sleeping duration per day?” Additionally, participants were instructed to answer using either “yes” or “no” responses to the following questions: “During the past year, has your daily life been burdened by depressed mood or feelings of hopelessness for more than two continuous weeks?”, “During the past year, did you ever seriously consider attempting suicide?”, “Have you ever made specific plans to commit suicide in the last year?”, and “Have you ever actually attempted suicide in the last year?” Perceived stress was evaluated using a four-point Likert scale (“excessive”, “much”, “often”, and “rarely”) for the question, “How stressed are you on a daily basis?” 

For statistical analysis, the level of stress was divided into two groups; those who answered “excessive”, “much”, and “often” were categorized as “yes” and those who answered “rarely” were categorized as “no”. Counseling experience was assessed using the “Yes/No” question, “Have you ever attended counseling, either face-to-face, via telephone, or through the Internet because of a mental health concern in the last year?” Quality of life (QoL) was assessed using the EuroQol-5 Dimension (EQ-5D) [27,28]. The EQ-5D is a questionnaire used to evaluate health-related QoL; it includes five dimensions: mobility, self-care, usual activities, pain/discomfort, and anxiety/depression. It comprises five questions, each of which is rated on a three-point Likert scale (ranging from “no”, to “severe–unable”). All questions were answered by the family members. 

Information collected for adolescent girls included sociodemographic factors (i.e., sex, age, socioeconomic status, and adolescent girls’ sleep duration) and mental health factors (i.e., stress, depressed mood, suicidal ideation, suicidal plans, and counseling experience). Information collected for parents included sociodemographic factors (i.e., sex, age, and education) and mental health factors (i.e., stress, depressed mood, suicidal ideation, suicidal plan, counseling experience, and QoL).

### 2.4. Statistical Analysis

To calculate representative estimates of the non-institutionalized Korean civilian population, all data of the adolescent girls who attempted suicide and their parents were assessed by applying weighted analyses. Weighted analyses were performed to compensate for the complex sampling design of KNHANES data and to allow for nationally representative prevalence estimates for the South Korean population [26]. 

First, adolescent girls were divided into two groups: suicidal attempts (suicidal attempt group) and others (non-suicidal attempt group). Considering the complex sampling design, the Rao-Scott Chi-Square test (categorical variables) and the general linear model (continuous variables) were used to evaluate the differences between sociodemographic and mental health characteristics related to suicide attempts among adolescent girls. The Rao-Scott Chi-Square test is similar to the Pearson Chi-Square test; however, the latter is not appropriate for a complex sampling design, as the data do not satisfy the required condition for the Pearson Chi-Square test [29]. Moreover, to compare the mean of continuous variables like sleep duration and QoL between the two groups, we used a t-test when considering the sampling weight.

Additionally, sociodemographic and mental health characteristics of parents were assessed using the Rao-Scott Chi-Square test and t-test described before. Second, the Chi-Square test was used to identify differences in the suicidal behaviors of adolescents related to their mothers’ and fathers’ depressive symptoms and suicidal behaviors. Third, hierarchical logistic regression was used to analyze risk factors for suicide attempts. We assessed adolescent girls’ sociodemographic and risk factors in Step 1 and evaluated the effect of maternal risk factors in Step 2. Also, to check multicollinearity, we evaluated VIF and check whether the value of VIF of each variable exceeded 5 [30]

Odds ratios (*OR*s) and 95% confidence intervals (*CI*s) were calculated for each factor. All statistical analyses were performed using SPSS version 27.0 (IBM SPSS Statistics for Windows, Armonk, NY, USA: IBM Corp). A *p*-value less than 0.05 (typically ≤ 0.05) was considered statistically significant.

## 3. Results

### 3.1. Sociodemographic and Mental Health Characteristics Related to Suicidal Attempts among Adolescent Girls

As shown in Table 1, 13 (1.46%) and 877 (98.54%) adolescent girls were classified into the suicide attempt group and non-suicidal attempt group, respectively. The findings revealed no significant differences between the two groups by age, stress, and socioeconomic status. The suicidal attempt group was significantly more likely to have experienced depressed mood (45.8% vs. 8.0%; *p* < 0.01), suicidal ideation (94.2% vs. 3.8%; *p* < 0.01), suicidal planning (48.9% vs. 0.5%; *p* < 0.01), and counseling experience (57.5% vs. 5.3%; *p* < 0.01). Moreover, the suicidal attempt group had a significantly shorter sleep duration than the non-suicidal attempt group (6.62 h vs. 7.86 h; *p* = 0.01) (Table 1).

### 3.2. Sociodemographic and Mental Health Characteristics Related to Parents of Adolescent Girls Who Attempted Suicide

Table 2 shows the sociodemographic and mental health characteristics of the parents of adolescent girls, according to suicide attempts. The findings revealed that maternal age, paternal counseling experience, education level, stress, and parents’ QoL were not significantly different between the two groups. Moreover, the suicidal attempt group was significantly more likely to have maternal counseling experience (29.2% vs. 3.5%; *p* < 0.01) and higher paternal age (51.88 ± 1.83 vs. 47.42 ± 0.19; *p* = 0.02; Table 2).

### 3.3. Difference in Adolescent Suicidal Behaviors Depending on Parental Suicidal Behaviors

Suicidal ideation in adolescent girls showed significant differences depending on maternal suicidal plans and attempts (*p* < 0.01; *p* < 0.01, respectively) (Table 3). Suicidal plans of adolescent girls showed significant differences depending on maternal suicidal ideation, plans, and attempts (*p* = 0.01, *p* < 0.01, *and p* < 0.01, respectively). Suicidal attempts by adolescent girls showed significant differences according to maternal suicidal plans and attempts (*p* < 0.01; *p* < 0.01, respectively). However, suicidal behaviors of adolescent girls did not significantly differ depending on paternal suicidal behaviors (Table 3).

### 3.4. Hierarchical Logistic Regression of Adolescent Girls’ Suicidal Attempts 

Hierarchical logistic regression analysis was conducted to analyze age, socioeconomic status, sleep duration, depressed mood, suicidal ideation, and suicidal plans among adolescent girls. As shown in Table 4, suicidal ideation (*OR* = 173.01) and suicidal plans (*OR* = 8.37) of adolescent girls were statistically significant explanatory variables for suicide attempts in Step 1. The model correctly identified 62.30% of adolescent girls who attempted suicide, and the model’s McFadden’s pseudo $R^{2}$ was 0.67. The four risk factors: maternal stress, depressed mood, suicidal plans, and suicide attempts, were added in Step 2. The model explained that suicidal ideation in adolescent girls (*OR* = 292.92) and suicidal plans and attempts of their mothers (*OR* = 6.39; *OR* =12.38, respectively) were significant risk factors for suicide attempts. The model correctly identified 53.40% of the participants with suicidal ideation, and the model’s McFadden’s pseudo-$R^{2}$ was 0.62. In Step 2, adding the parental risk factor to the model increased the classification accuracy of adolescent girls’ suicide attempts by 8.9% and the McFadden’s pseudo *R*^2^ by 0.04.

## 4. Discussion

We assessed factors related to suicidal attempts among adolescent girls and the relationship between adolescents’ suicide attempts, their parents’ mental health and suicidal behaviors. This study had two key findings. First, suicidal ideation in adolescent girls was a risk factor for suicide attempts. Second, suicidal plans and attempts in adolescent girls’ mothers were important risk factors for suicide attempts among adolescent girls.

In general, since suicidal ideation precedes suicidal attempts, a natural outcome can be observed between the two variables. However, suicidal ideation reflects a transient state and can change continuously [8]. As suicidal ideation was highly related to suicidal attempts in our study, we reconfirmed the importance of evaluating suicidal ideation. Adolescent girls who had attempted suicide reported higher depressed mood, suicidal ideation, suicidal planning, and counseling experience; moreover, they had shorter sleep durations than adolescent girls who had not attempted suicide. Previous studies also support our findings. Adolescents with suicidal behaviors experienced more severe levels of depression [13,18,31], while those who had attempted suicide had more depressive symptoms than those who did not attempt suicide [9]. Moreover, sleep satisfaction was lower in adolescents who had attempted suicide than in those who had not attempted suicide [7,9,32]. The National Sleep Foundation (NSF) [33] recommends a sleep duration of 8–10 h each night for adolescents. Moreover, it does not recommend less than seven hours or more than 11 h of sleep. Our results reveal that the average sleep duration of adolescent girls who attempted suicide was 6.62 h, which is insufficient, based on the NSF recommendation. Meanwhile, 15.4% of adolescent girls who had attempted suicide had visited mental health professionals. Although they had significantly more experience with mental health professionals than adolescent girls who had not attempted suicide, the rate of therapeutic interventions is insufficient. This is because the current KNHANES does not involve treatment by psychiatrists or related mental health experts, but simply evaluates whether one has had more than one experience of psychological counseling. Further efforts are needed to promote appropriate therapeutic interventions for adolescents.

We also found that suicidal plans and attempts in adolescent girls’ mothers were important risk factors for suicide attempts among adolescent girls. Previous studies [8,17,31,34,35] have shown that suicidal behaviors are affected by familial transmission and occur independently of the transmission of mood disorders. Studies report that among suicidal behaviors, suicidal attempts are transmitted more than suicidal ideation [36,37]. Moreover, several studies have consistently reported that suicide attempt risk in the offspring of people who attempted suicide increases by approximately five-fold [31,34,35]. Brent et al. [31] reported that impulsive aggression and mood disorders play an important role in increasing offspring suicide attempts. In particular, since mood disorders are an important factor of early onset suicidal behavior among youth, prevention and treatment of mood disorders to avoid suicide are warranted. While Lieb et al.’s [37] study did not evaluate paternal suicidal behaviors, it revealed that the rate of suicide attempts in offspring increased by nine times when there was a maternal suicide attempt. In previous studies, parental mental health was reported to have a greater effect on adolescents of the same gender [22,23,24]. Moreover, adolescent girls are more affected by maternal than paternal mental health because mothers generally feel more responsible than fathers for parenting, and adolescent girls have more emotional interactions with mothers than their fathers [21]. These previous results support our findings. When the mother had a suicidal plan or attempt, there was a significantly higher rate of suicide plans and attempts among adolescent girls; however, no difference was found in the rate of suicidal behavior among adolescent girls according to the suicidal behavior of their fathers. In our study, the small number of paternal suicidal behaviors among adolescent girls with suicidal behaviors limited our interpretation of the results. In future studies, larger sample sizes with long-term data will be needed to supplement our findings.

The limitations of our study are as follows: first, this study did not evaluate the number of suicide attempts or the severity of suicide intention among adolescent girls and parents. Second, this study’s cross-sectional design limited the understanding of the causal relationship between suicide attempts of adolescent girls and maternal suicidal behaviors.

## 5. Conclusions

Despite the limitations, our study found that suicidal plans and attempts in adolescent girls’ mothers were important risk factors for suicide attempts among adolescent girls. Furthermore, we found that paternal suicidal behaviors were unrelated to suicidal behaviors among adolescent girls. Prospective studies should explore specific methods for effective prevention and treatment through path analysis of the related factors affecting suicidal behavior of adolescents according to their parents’ gender.

## Figures and Tables

**Table 1 ijerph-18-08122-t001:** Sociodemographic and mental health characteristics related to suicidal attempts among adolescent girls.

Suicidal Attempt				
		Yes (*n* = 13, 1.46%)	No (*n* = 877, 98.54%)	
		Mean ± *SE*	Mean ± *SE*	*p*
Adolescents	Age (years)			
	12–15	14.05 ± 0.46	13.6 ± 0.05	0.32
	16–18	16.77 ± 0.27	16.99 ± 0.05	0.43
	Stress	100% (0.0)	85.9% (1.3)	0.18
	Depressed mood	45.8% (15.2)	8.0% (0.9)	<0.01
	Suicidal ideation	94.2% (5.7)	3.8% (0.7)	<0.01
	Suicidal plan	48.9% (15.7)	0.5% (0.2)	<0.01
	Sleep duration (hours)	6.62 ± 0.37	7.87 ± 0.1	<0.01
	Counseling experience	57.5% (15.4)	5.3% (0.7)	<0.01
	Socioeconomic status			
	low	30.0% (15.6)	10.6% (1.2)	0.14
	middle	37.5% (14.2)	57.3% (2.2)	
	high	32.5% (15.0)	32.1% (2.3)	

Note. *SE* = standard error.

**Table 2 ijerph-18-08122-t002:** Sociodemographic and mental health characteristics related to parents of adolescent girls who attempted suicide.

		Suicidal Attempt		
		Yes (*n* = 13, 2.1%)	No (*n* = 842, 97.9%)	
		Mean ± *SE*	Mean ± *SE*	*p*
Mother	Age (years)	46.72 ± 2.24	44.51 ± 0.17	0.32
	Education			
	High school graduation or less	57.0% (15.4)	47.5% (2.1)	0.55
	University graduate or higher	43.0% (15.4)	52.5% (2.1)	
	Stress	94.3% (5.6)	91.4% (1.0)	0.67
	Depressed mood	33.5% (13.9)	44.3% (2.5)	0.47
	Counseling experience	29.2% (15.5)	3.5% (0.7)	<0.01
	Quality of life	0.972 ± 0.01	0.969 ± 0.0	0.82
		Suicidal attempt		
		Yes (*n* = 7, 1.6%)	No (*n* = 683, 98.4%)	*p*
Father	Age (years)	51.88 ± 1.83	47.42 ± 0.19	0.02
	Education			
	High school graduation or less	67.5% (20.3)	42.3% (2.4)	0.24
	University graduate or higher	32.5% (20.3)	57.7% (2.4)	
	Stress	82.1% (12.5)	90.1% (1.3)	0.42
	Depressed mood	11.4% (11.0)	47.4% (2.8)	0.04
	Counseling experience	0% (0.0)	0.6% (0.3)	0.93
	Quality of life	0.95 ± 0.04	0.98 ± 0.0	0.39

Note. *SE* = standard error.

**Table 3 ijerph-18-08122-t003:** Difference in adolescent suicidal behaviors depending on parental suicidal behaviors.

Adolescent Girls										
		Suicidal ideation			Suicidal plan			Suicidal attempt		
		Yes % (*SE*)	No % (*SE*)	*p*	Yes % (*SE*)	No % (*SE*)	*p*	Yes % (*SE*)	No % (*SE*)	*p*
Mothers	n (%)	46 (5.4%)	809 (94.6%)		13 (1.5%)	842 (98.5%)		13 (2.1%)	842 (97.9%)	
	Suicidal ideation	6.0% (3.6)	3.3% (0.8)	0.15	21.5% (12.2)	3.2% (0.7)	0.01	7.2% (7.1)	3.3% (0.8)	0.44
	Suicidal plan	2.8% (2.8)	0.6% (0.3)	<0.01	10.2% (9.7)	0.6% (0.3)	<0.01	7.2% (7.1)	0.6% (0.3)	<0.01
	Suicidal attempt	2.8% (2.8)	0.1% (0.1)	<0.01	10.2% (9.7)	0.1% (0.1)	<0.01	7.2% (7.1)	0.1% (0.1)	<0.01
Fathers	n (%)	29 (5.0%)	616 (95%)		4 (0.6%)	641 (99.4%)		7 (1.6%)	638 (98.4%)	
	Suicidal ideation	0% (0.0)	2.2% (0.7)	0.45	0% (0.0)	2.1% (0.6)	0.91	0% (0.0)	2.2% (0.6)	0.7
	Suicidal plan	0% (0.0)	0.3% (0.2)	0.84	0% (0.0)	0.3% (0.2)	0.98	0% (0.0)	0.3% (0.2)	0.86
	Suicidal attempt	0% (0.0)	0.3% (0.2)	0.84	0% (0.0)	0.3% (0.2)	0.98	0% (0.0)	0.3% (0.2)	0.95

Note. *SE* = standard error.

**Table 4 ijerph-18-08122-t004:** Hierarchical logistic regression of adolescent girls’ suicidal attempts.

Variables			Step 1			Step 2		
			*OR*	95% *CI*	VIF	*OR*	95% *CI*	VIF
Demographic	Adolescents	Age	0.82	0.58–1.18	1.68	0.82	0.55–1.22	2.83
		SES (low) ^a^	0.78	0.16–3.86	1.57	0.44	0.06–3.10	2.47
Risk factors	Adolescents	Sleep duration	0.58	0.27–3.86	2.37	0.54	0.21–1.39	3.70
		Depressed mood	1.38	0.19–9.93	2.82	1.64	0.23–11.74	3.16
		Suicidal ideation	173.01 ***	10.68–2802.98	3.16	292.92 ***	21.41–4062.14	3.93
		Suicidal plan	8.37 **	1.54–45.48	1.31	5.79	0.76–44.164	1.76
	Mothers	Stress				2.21	0.22–22.25	1.51
		Depressed mood				0.30	0.06–1.55	2.27
		Suicidal plan				6.39 *	1.25–32.75	1.76
		Suicidal attempt				12.38 *	1.56–98.26	2.03

Note. ^a^ Reference group: Middle and high socioeconomic groups *OR* = odds ratio; *CI* = confidence interval; SES = Socioeconomic status; * *p* < 0.05; ** *p* < 0.01; *** *p*< 0.001.

## Data Availability

The datasets used and/or analyzed during the current study are available from https://knhanes.kdca.go.kr/knhanes/sub03/sub03_02_05.do (Korean Centers for Disease Control and Prevention. *Korea National Health and Nutrition Examination Survey (I–IV) 2019)*.

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
