# Peer review of "Parental Factors Related to Adolescent Girls’ Suicide Attempts: A Cross-Sectional Study from 2015 to 2018"

_ijerph, 2021, doi:10.3390/ijerph18158122_

Round 1

Reviewer 1 Report

This is an excellent article and the authors provide substantive data and research to support their findings.  The only paragraph I found "odd" was at the very end--the one preceding the last paragraph.  It begins "Authors should discuss the results...."  That paragraph did not seem to fit with the rest of the paper.  Other than that, this is an excellent paper and I hope the editors will publish it.

Reviewer 2 Report

It is a very interesting paper dealing with the challenging issue of adolescent suicidality and concentrating on the female suicide attempters. The paper is novel in the sense that female adolescent suicidality is usually overlooked since  epidemiological research on suicide focuses on the highest frequency of male suicidality.

Beyond self injurious acts that are mentioned in the text, it would be of interest to add any differences in the methods used by female adolescents in contrast to their male counterparts, since it is well known from literature that male attempts and suicides are usually more violent.

Also, the finding of suicidal ideation preceding the attempt  sounds quite self-explanatory, unless it provides evidence that suicidal ideation (ranging from passive to active) is of equal importance as suicidal planning in terms of dangerousness. I would like to see a clarification of the finding.

Finally, the finding of maternal suicide plans and attempts influencing their daughters suicidal behavior in terms of vicarious learning is really important and fascinating.

Reviewer 3 Report

1) The author noticed only the dynamics of suicide mortality in South Korea between 2017 to 2018 (L30~34). Authors should clarify whether that this dynamic is an atypical transient or a long-term trend in South Korea. Reviewer know the increasing suicide trends of suicide mortality of female in South Korea, but reviewer feels that the author's statement may be too exaggerated to give readers an realistic impression. Authors need to elaborate on trends in suicide mortality, at least during the study period 2015-2018. If author describe the trends of suicide mortality during a decade, i.e. 2009~2018, majority of reader can understand the trends in South Korea. If author can agree with my suggestion, please add any data of suicide mortalities of teens male, female and their parents generations results section.

2) Please add some reference regarding ‘internal individual factors and external environmental factors’ (L35-36)

3) I disagree with considering suicide as a mental illness/disorder. Therefore, please improve the word “Parental psychopathology, such as suicidal behaviors” (L55) and other parts.

4) Please describe the suicide mortalities of generations of mother and father during 2009~2018. The author’s is hypothesis is not able to appropriate, if the suicide mortality or attempted suicide rate of the mother generation has not increased or not relate. Reviewer appreciate the importance of this manuscript, but if the hypothesis is not reasonable, author should improve the hypothesis to be a more rational. Therefore, reviewer request again the detailed data about suicide mortalities of teens and their parents generation during 2009~2018.

5) Reviewer did try to obtain suicide data from KDCA based on the author's description, but I could not reach to the database. If the suicide data in KDCA is public, please authors describe the correct web address.

6) EQ-5D battery has been established. Are the stress and depression questions based on past medical questionnaires? Additionally, is control data published that can be compared with healthy Korean population?

7) Did author adopt general linear model for analysis sleep time? The general linear model is vague. Please specify a more appropriate statistical analysis method model name. Did you perform a Robustness analysis? Especially, did author clarify the variance inflation factor?

8) Reviewer cannot understand the statistical methods. Please describe statistical analysis data with the statistical methods in the results section.

Round 2

Reviewer 3 Report

Thank you for the polite improvement of the authors.
However, reviewer cannot agree with a part of responses of authors in the revised version.
Reviewer disagree with describing the statistics methods according to the menu of SPSS.
In the response 7 to reviewer, “Strictly speaking, it is a linear regression model”…….
Reviewer evaluated the statistical method, “complex samples general linear model (CSGLM )” was critically wrong.
Furthermore, “Moreover, the variance inflation factor (VIF) showed no evidence of multicollinearity.” Reviewer cannot agree with the response comment. If authors analysed using linear regression analysis, the comments of reviewer regarding VIF was untargeted comments; however, if the authors analysed using CSGLM, author should describe the criteria of VIF and the VIF real numbers. In conclusion, reviewer evaluated that authors deprecates statistical analysis. Reviewer strongly request more carefully improvement about statistical analysis.

Author Response

And we would like to thank Editage (www.editage.co.kr) for English language editing.
